# Measuring the Effectiveness of Food Policy Councils in Major Cities in the United States

**DOI:** 10.3390/foods12091854

**Published:** 2023-04-29

**Authors:** Camille Range, Sabine O’Hara, Tia Jeffery, Etienne C. Toussaint

**Affiliations:** 1College of Agriculture, Urban Sustainability and Environmental Sciences (CAUSES), University of the District of Columbia (UDC), Washington, DC 20008, USA; 2School of Law, University of South Carolina, 1525 Senate Street, Columbia, SC 29208, USA

**Keywords:** urban food systems, food supply and demand, food policy, leadership, diversity and inclusion

## Abstract

The United States food system is comprised of a diversity of stakeholders representing a range of sectors including agriculture, health, hospitality, and other sectors of the economy. Coordinating these wide-ranging aspects of the food system is a challenging responsibility of the United States Department of Agriculture (USDA) and its counterparts at the state and county level. While the USDA acknowledges the institutional importance of diversity, it has only recently begun to meaningfully consider the agricultural needs of urban environments, the unique experiences from communities of color, and opportunities for urban agriculture in its purview. This study provides a framework for food systems leaders, specifically in food policy councils (FPCs) in large urban communities, to assess the visibility and effectiveness of their diversity and inclusion initiatives represented across three key domains: (1) leadership and governance structures, (2) key stakeholder engagement strategies, and (3) advance food justice. A cluster analysis of the study results for FPCs in 19 U.S. cities and metropolitan areas across these three domains revealed four distinct groups of food policy councils. Results from the assessment of the urban FPCs reveal that they succeed in embedding diversity and inclusion in all three diversity measures, with key stakeholder engagement receiving the highest score at 73%. On average, FPCs received a score of 54% when assessing diversity and inclusion in their leadership and governance and a score of 49% in their activities to advance food justice. Results of the analysis also highlights opportunities to improve individual, thematic, and cumulative scores. Participation from food systems leadership groups such as food policy councils will be vital in advocating for equitable food systems through urban producer and consumer programs in the upcoming reauthorization of the U.S. farm bill in 2024.

## 1. Introduction

Since the early 20th century, the United States Department of Agriculture (USDA) has played a critical leadership role in coordinating the interests of various stakeholders across several food-related sectors, including agriculture, health, hospitality, manufacturing, technology, and petrochemicals [1]. Institutional efforts to uplift the importance of diversity and inclusion among these stakeholders, in relation to their engagement with the United States (U.S.) food system, has historically reflected a wide range of perspectives, spanning the views of both producers and consumers. At the same time, the USDA readily acknowledges the agency’s challenging history of racial discrimination and race-based exclusion, which has fallen short of “providing equitable service to Tribal Nations, communities of color, rural, and other underserved and underrepresented persons” [2]. These shortcomings not only implicate the experiences of U.S. food producers, but also of food consumers and the nearly 100,000 USDA employees who support the U.S. food system across the country [2].

### 1.1. Importance of Diversity, Equity and Inclusion in the U.S. Food System

Diversity is recognized as a vital component of organizational effectiveness and sustainability of social and environmental systems [3,4,5,6,7,8]. The diversity of organizational member demographics (e.g., diversity of race, age, gender, religious affiliations, etc.) serves as a conduit for the engagement of a broad range of ideas, experiences, and skills that each contribute toward a social system’s or organization’s ability to consider and implement shared goals [5,8,9]. Disparate experiences of health disparity, economic inequality, and environmental racism have led the way in the United States diversity debate and have served as drivers of social change. For example, the American Public Health Association and over 200 government offices, leaders, and educational institutions declared racism as a public health crisis in August of 2021 [10]. Similar trends to adopt an equity- and diversity-centered framework to support organizational change have emerged in government agencies, municipal administrations, nonprofit and for-profit organizations have been on the rise [10,11,12].

The food system is a relative newcomer to the diversity debate in the U.S. Research highlights the importance of adopting various principles of diversity, inclusive leadership principles, equity, and access to eradicate disparities in the current food system [11,12,13,14]. Recently, the USDA made a significant commitment to rectify past neglect of diversity, equity, and inclusion (DEI) by developing a Diversity, Equity, Inclusion and Accessibility Strategic Plan, and hiring its first Chief Diversity and Inclusion Officer [2]. These internal DEI strategies seek to bolster institutional knowledge and establish external-facing policies that will support communities of color and other underserved and underrepresented groups that the USDA serves [2]. This focus is timely, as the 2024 omnibus farm bill promises a new opportunity to revisit current food systems leadership strategies and policies designed to enhance food and nutrition security, community health outcomes, and environmental sustainability [15]. Furthermore, research suggests that improved diversity and inclusion in food systems leadership is an important element of successful long-term municipal planning and policy objectives [16,17,18,19].

This study examines the representation of diversity and inclusion in food systems focused institutions that are located in densely populated urban and metropolitan areas across the United States. Urban communities in the United States have historically been characterized by higher levels of diversity than rural areas [20]. Further, most food consumers live in cities [20,21]; however, food producers have been virtually absent from cities and metropolitan areas. Indeed, it was not until the most recent farm bill of 2018 that urban agriculture was even mentioned [22]. To amplify the food systems challenges facing U.S. urban and metropolitan communities, we provide a diversity and inclusion framework for food and agriculture industry leaders, lawmakers, and advocates in food systems focused institutions in U.S. cities to assess the effectiveness of their DEI goals and objectives.

### 1.2. U.S. Food Policy—Bridging Production and Consumption

Historically, the U.S. food system has primarily produced commodity crops, including corn, soybeans, cattle, poultry and eggs, and dairy products [23]. In 2017, these five commodities—primarily produced by the Midwest and Southern regions—accounted for 66% ($255B) of total U.S. agriculture sales [24]. Additionally, these regions have served as the home to 23 of the 32 (72%) USDA secretaries since the agency’s founding in 1862 [25,26]. As a result, the programmatic focus of the USDA has historically reflected the preferences, needs, and interests of large farms and other agricultural producers from rural areas in the Midwest and Southern regions. This lack of diversity and inclusion in the agency’s institutional focus has invariably impacted the portrayal of the U.S. food system and the resulting policy priorities at both the federal and state level.

However, the breadth of stakeholders across the U.S. food system, each with diverse and often competing interests, has resulted in the negotiated inclusion of consumer-focused policies—chiefly, food subsidies—in the U.S. Farm Bill since at least the 1920s [27]. The farm bill is reauthorized every five years with amendments that reflect political debates about these diverse interests [27]. For example, in the most recent 2018 Farm Bill, only 24% of funding was allocated to support producer-focused programs through loans, crop insurance, energy, conservation, and research. Since, in recent years, consumer-focused programs have constituted at least 76% ($326B) of USDA spending, critics question the sufficiency of the $102B allocated for producer-focused programs [22]. Support for the U.S. consumer was historically accelerated by federal nutrition assistance advocacy designed to fight food insecurity [28]. As a result, a large percentage of the farm bill’s funding allocations currently support the Supplemental Nutrition Assistance Program (SNAP)—formerly known as the food stamp program—alongside the Women Infant and Children (WIC) program, school food programs, senior farmers market programs, and emergency food assistance [22]. SNAP is currently the largest U.S. federal nutrition assistance program, accounting for 65% of total USDA nutrition title expenditures in FY 2020 [29]. The budgetary shift from producer to consumer programs not only highlights the evolution of the USDA’s programmatic focus away from its original audience of food producers, but it also underscores the modern lived experience of many food consumers in the U.S. who exist in a state of food insecurity.

This relatively recent shift has inspired the USDA to investigate why federal food and nutrition have failed to combat persistent hunger and food insecurity in the United States, particularly in communities of color that have been historically underserved and underrepresented in the U.S. food system. SNAP’s typical user experience is associated with densely populated urban environments where the prevalence of food insecurity abounds. As of 2021, over 80% of SNAP users lived in urban or metropolitan communities and reflected the demographic diversity of such urban communities across America: 36.5% white, 25.8% black, 16% Hispanic, 3% Asian, 1.5% Native American, 0.8% multiple races, and 16% identifying as race unknown [21]. Accordingly, while food insecurity is a national challenge, it is most prevalent among urban-based food consumers—particularly people of color—who also disproportionately experience insecurity in employment, housing, environmental health, and health care [30,31,32]. Indeed, in 2021, there were six times more households experiencing food insecurity in urban or metropolitan areas compared to rural areas [33]. Scholars have linked the lack of access to healthy and nutrient-rich food in predominantly low-income and non-white urban communities to their high rates of chronic diet-related diseases, such as obesity, diabetes, and cardiovascular disease [30,32,34].

### 1.3. Building Inclusive and Effective Food Systems Leadership

Institutional leadership across the U.S. food system remains largely representative of rural communities, underscoring the persistence of barriers to diversity, equity, access, and inclusion in the U.S. food system. While the USDA primarily distributes farm bill funds to state and county-level offices, cities often lack representation among institutional leaders in the food and agriculture sector [35]. Recently, this lack of representation has inspired calls from urban-based producers and community-based advocates for the food industry to embrace concepts like “food democracy” and “food sovereignty” [16,28,36,37,38]. Moreover, food justice advocates have promoted consumer-driven strategies that demand more localized food production systems to empower local producers, boost positive environmental outcomes, and increase resilience [16,39,40]. However, accounting for the wide-ranging impacts of both consumer-oriented and producer-oriented food programs that prioritize fruit and vegetable production can be challenging in the absence of an urban representation among food and agriculture institutional leaders [41]. There is therefore a continued need for food and agriculture institutions to foster inclusive perspectives and experiences, especially in urban communities.

Food policy councils (FPCs) are recent additions to the urban food systems landscape intended to fill the void of urban representation in food and agriculture. An FPC is defined as an organized group of stakeholders working to address food systems issues and needs at the local, state, regional or tribal nation level [38,42]. A review of U.S. FPCs reveals that they are diverse in terms of their organizational structure, stakeholder representation, and relationship with local municipalities [16,35,36,37,42]. A survey of FPCs conducted by The John Hopkins Center for a Livable Future collected descriptive data, including the locale representation, organizational structure, priorities, and activities of FPCs [37,38,42]. However, the survey did not include specific questions regarding the membership selection process, size, representation, and decision-making processes of the FPCs [37,38,42]. The majority of FPCs were established after 2010. Research on their institutional characteristics and effectiveness is therefore limited in its temporal scope [42]. Moreover, the recent COVID-19 pandemic transformed the role of FPCs to support the emergency food response of municipalities during the pandemic [43,44]. This adaptation further challenges an analysis of the organizational focus of the FPCs [16,37,43,45].

Despite these challenges, a relatively consistent role of the FPC emerges. They are chiefly viewed as advisors on local governance and decision-making processes related to the food systems priorities of diverse stakeholders [37,38,46]. FPCs are also frequently viewed as catalysts for creating and maintaining collaborations between municipal governments and other stakeholders in support of at-risk communities, including the effective collaboration between local and regional food producers and food consumers [18,21,26,27]. Given these challenging bridge-building roles, the effectiveness of FPCs requires their ability to successfully navigate the diversity and inclusion challenges faced by food systems stakeholders in urban communities [16,43,44,47]. This study assesses how effective FPCs have been in embedding principles of diversity and inclusion into their (1) leadership and governance structures, (2) stakeholder engagement processes, and (3) programmatic activities in pursuit of a more equitable food system. By utilizing publicly available information, we examined key diversity and inclusion characteristics of food policy councils in the 25 largest cities in the United States. These 25 cities and their FPCs represent 10% of the total U.S. population [20]. Our data sources include the websites of the FPCs in our sample, as well as other published resources. Since FPC websites also serve as key communication tools for food industry stakeholders, our study makes transparent the food equity and justice priorities reflected in the activities of the FPCs in our sample.

## 2. Materials and Methods

### 2.1. Analyzing Food Policy Councils (FPCs)

FPCs are generally comprised of diverse stakeholders working to address food systems issues and establish food policy priorities at the local, state, and regional levels, or within tribal nations [38,42]. Currently, there are 195 active FPCs in the United States (U.S.) and three in tribal nations [42]. The majority of the FPCs are represented at the county-level. Their average age is seven years [42]. The John Hopkins Center for a Livable Future and its Food Policy Network serves as a data hub for FPCs across the United States [42,43]. It also conducts an annual assessment of FPC leadership and activities. Despite this growing body of data, many information gaps remain [16,42,43,44,47]. The most vulnerable stakeholders of FPCs are food insecure populations. These groups are also predominantly low-income and non-white, which comprise the largest group of food assistance recipients [21]. Given the recent increases in government funding for urban agriculture—which opens new opportunities for local governments to meet urban-based consumer needs—examining the effectiveness of FPCs remains an important priority. The drafters of the 2018 Farm Bill acknowledged the dire need for positive changes in the U.S. food system. It also established the Office of Urban Agriculture, alongside 17 urban agriculture and innovation committees in 17 U.S. cities and metropolitan areas nationwide [48]. This new focus recognizes the potential for urban agriculture to benefit cities, including enhancing access to fresh produce, boosting health and wellness, and strengthening the local economy in a sustainable manner [49,50]. Urban agriculture also offers a unique opportunity to support diverse food producers and consumers of color as cities strive to advance racial equity and food justice at the municipal-level.

The 25 U.S. cities included in this study were selected based on data from the 2020 U.S. Census [20]. Figure 1 highlights the selected cities and metro areas, and it identifies active or inactive FPCs. Active FPCs were identified by the existence of a live web address and related documents that confirm recent organizational activity as of January 2023. In addition to the web search, the selected cities were cross referenced with The Johns Hopkins Center for a Livable Future’s Food Policy Network database to identify the associated FPC and its designated websites and other publicly available information. The publicly available information for each FPC includes its inauguration date, meeting minutes, reports, program assessments, and strategic and action plans. Similar content analysis of FPCs based on publicly available information has been previously undertaken for a smaller sample of FPCs [35,37].

FPCs are commonly categorized by their organization type. For example, an FPC may operate as a government office or taskforce, as a government-affiliated advisory council, as a nonprofit organization, or as a coalition [42,46]. Operation as a nonprofit organization is the most common organizational structure, comprising 48% of FPCs. Alternatively, 25% are structured as a government entity, 20% are formed by grassroots coalitions, 5% are embedded within a university, and 2% utilize another organizational structure [42]. For this study, the organizational affiliation at the time of data collection (January 2023) was used for FPC categorization. The majority of the urban FPCs in our sample are government-affiliated advisory councils (56%) and nonprofit organizations (32%). There were no coalition-based FPCs represented in this study. In addition, three of the cities in our sample (New York, NY, USA; Phoenix, AZ, USA; Seattle, WA, USA) contain FPCs that operate as a government office or intergovernmental task force.

Since our sample of FPCs was taken from larger cities and metropolitan areas across the United States, it includes a higher representation of city and municipality based FPCs (44%) as compared to the reported national average of 12% [42]. Our sample also includes a higher representation of FPCs structured as city–county partnerships (28%) as compared to the reported national average of 19% [42]. Further, while 37% of FPCs in the United States are organized at the county level, only 16% of the FPCs in our sample pool operate at the county level. Therefore, we expect the FPCs in our sample to reflect the heightened awareness of DEI-related issues that are frequently associated with diverse urban environments. Descriptive characteristics included in our study include the organizational structure of the FPC, the pathway to their initial formation, local stakeholder representation, and the U.S. region where the FPC is located [51].

### 2.2. A Framework for Assessing Food Policy Councils

A review of the literature reveals three major themes regarding the effectiveness of FPCs, including: (1) leadership and governance structure, (2) key stakeholder engagement, and (3) activities toward food justice (Figure 2). We apply these three themes to our sample of FPCs by analyzing publicly available information for each FPC. A secondary qualitative content analysis was undertaken to further define and rank criteria in all three themes for each FPC in our sample. A small number of previous studies used a similar methodology of analyzing publicly available information [52,53,54]. While current literature contributes to sharing best practices and lessons learned, FPCs across the United States struggle to define and track indicators for success [36,38,45,47]. Calancie et al. [55] validated a self-assessment tool for FPC members to assess their strengths and areas of improvement, which can then be compared to others over time. Their tool provides opportunities to identify themes (organizational capacity, social capital, and council effectiveness) that are aligned with identified community outcomes [55,56]. The current study aims to contribute to the existing literature by providing an alternative methodology for FPCs to enhance the visibility of similar themes relevant to DEI concerns, including leadership and governance structure, key stakeholder engagement, and activities toward food justice. To rank the contributions of the individual criteria included in our three themes, we adopted a rating score from 0 to 2 in order to assess leadership and governance structure, key stakeholder engagement, and activities advancing food justice for each urban FPC. The rubric definitions are based on an assessment of publicly available information from each FPC, which are included Appendix A. Individual scores for each criterion within each theme were used to calculate the total score by theme and for comparison across FPCs. Results from our calculations were then translated into a percentage of the maximally achievable score for each of the criteria within each of the three themes.

While previous studies have measured the effectiveness of FPCs based on the degree of systems level thinking and leadership, our study makes a unique contribution by examining the effectiveness of FPCs as demonstrated by their diversity, equity, inclusion, and access of the councils [15,16,37,46]. One of the previous studies, which used a similar methodological approach based on the three themes summarized in Figure 1, defined the leadership and governance (LG) theme as including local municipal governance relations as an influential criterion (LG.1). There is evidence to support the value of FPCs having access to municipal government resources, relationships, and coordination functions [35,36,37]. Based on these previous findings, our study assigns a score of 2 to those FPCs that show strong municipal government relations represented by a government-affiliated or government-sanctioned organizational model. On the contrary, local political dynamics have been cited as a deterrent for the effectiveness of FPCs [16]. We also consider the degree of structural autonomy of the FPC, including whether it is sponsored by or represented by a nonprofit organizational structure, such as the 501(c)(3) nonprofit incorporation structure typically utilized by U.S. nonprofit organizations. If the FPC includes local municipal representation in its leadership and meetings, it is assigned a score of 1. Absence of criteria information, or the inapplicability of criteria to the FPC, resulted in a score of 0.

In our data collection, we determined that governance documents, such as bylaws (LG.2), typically detail the relationship that FPCs maintain with their leadership organizational structure (LG.3). We therefore analyze whether FPC bylaws embed DEI-related requirements for their leadership. FPCs that incorporated DEI-related metrics into their bylaws regarding their leadership organizational structure were assigned the highest score of 2. FPCs that exhibited only some evidence of the respective criteria received a score of 1. Absence of any relevant information, or observation that the criteria inapplicable to the FPC, resulted in a score of 0. It is noteworthy that some of the leadership and governance criteria were not applicable for a small number of the FPCs in our sample due to the nature of their organizational structure. For example, we observed that a designated food policy office within a municipal government was unlikely to maintain governance bylaws (LG.2) or leadership position details (LG.3) since these criteria are not applicable due to their organizational location within a larger government institution. Thus, in such cases, bylaws are generally unavailable.

In addition to the leadership and governance of the FPC, their effectiveness can be enhanced with the inclusion of more diverse groups of residents and other key food systems stakeholders [17,19,35,37]. One of the primary responsibilities of FPCs is to offer advice on local food systems related governance and decision-making processes. This implies that FPCs must not only advocate for vulnerable populations, but they must also ensure that the needs of such populations are reflected in the shared priorities of other key stakeholders [37,38,46]. For this study, key stakeholder engagement (KSE) is defined by several criteria, including the presence of contact and meeting information (KSE.1), modes of communication (KSE.2), and essential website content (KSE.3). FPCs with information regarding these criteria available on their website were assigned the highest score of 2, while the presence of some information of the respective criteria was assigned a score of 1. Absence of the relevant criteria information, or the non-existence of the criteria due to its inapplicability, resulted in the FPC being awarded a score of 0.

Lastly, research suggests that enhanced diversity and inclusion in food systems leadership benefits the long-term municipal planning and policy agenda of local governments to promote food equity and justice [16,17,18,55]. In this study, we use four criteria to define the activities that advance food justice. The first is a visible commitment to diversity and inclusion through a position/mission statement on the website or related materials (AFJ.1). Evidence of this information and the allocation of resources toward its implementation were assigned the highest score of 2. The presence of a position/mission statement on the website without additional evidence was assigned a score of 1. Lack of evidence of a written commitment to diversity and inclusion resulted in a score of 0. Remaining activities advancing food justice were measured by the representation of both urban producers and consumers in the respective FPCs’ goals and priorities through their designated working groups (AFJ.2), published reports (AFJ.3), and other available resources (AFJ.4). Representation for both urban producers and consumers was assigned the highest score of 2, while representation from only one was assigned a score of 1. Absence of resources for both urban consumers and producers resulted in a score of 0.

Using this rating system for each individual criterion, the total score and average thematic score was calculated for each FPC in our sample. The results are reported as a percentage of the maximum achievable score. A cluster analysis, using the average thematic score, then defines the effectiveness of a FPC as “high” or “low”. FPCs with a thematic score lower than the cumulative average thematic score are classified as “low”. Those with a score higher than the cumulative average thematic score are classified as “high”. From this classification, FPCs can be grouped based on their high or low effectiveness in leadership and governance, key stakeholder engagement and activities advancing food justice. The average thematic and total scores are then calculated and compared across the entire sample. The comparative results provide useful insight into urban food systems leadership based on our indicators of effectiveness for urban FPCs.

## 3. Results

Results from our analysis reveal that 22 of the 25 major U.S. cities we examined have an active FPC. There is no evidence of active food policy councils in Dallas, TX, USA, El Paso, TX, USA, and Oklahoma City, OK, USA. In addition, three of our identified FPCs appear to have been inactive at the time of data collection. They are the FPCs located in Jacksonville, FL, Nashville, TN, and Portland, OR. Information regarding these FPCs was retrieved from their archived websites and/or the published literature [57,58], but was ultimately excluded from the analysis.

### 3.1. Descriptive Results of Major Urban Cities and FPCs

Our secondary content analysis produced individual criteria, thematic and total scores for each city and its FPC as a percentage of the maximum achievable score. The average score for the selected urban FPCs (57%) was achieved from a maximum score achievable if all categories and subcategories had achieved the maximum score of 2 on our 0–2 rating scale. Of the 25 major U.S. cities, Washington, DC (The DC Food Policy Council) has the highest scoring FPC with a total of 22 points out of 22 achievable points resulting in a ratio of 100%. This suggests that the Washington, DC FPC is the most effective one in embedding diversity and inclusion in their leadership and governance structure (LG), their key stakeholder engagement (KSE), and their activities focused on advancing food justice (AFJ). The lowest ranking FPCs were located in Charlotte-Mecklenburg (NC), at 27%, and the City of Phoenix (AZ), at 28%, of the maximum achievable score. These results must, however, be interpreted as reflecting the performance of FPCs based on publicly available information.

Cumulative average scores were calculated as a percentage to compare results across descriptive categories (Table 1). On average, government-affiliated urban FPCs score higher (63%) than nonprofit-based urban FPCs (48%) in embedding diversity and inclusion into their LG, KSE, and AFJ themes. Similarly, urban FPCs established through the municipal government score higher (63%) than those established through a grant (48%) or a coalition (46%). Urban FPCs representing the city and county score higher (62%) than those only representing the city/municipality (56%) or county (50%) separately. While there is more representation from the West and South, urban FPCs in the Northeast (65%) and Midwest (59%) were more effective in embedding diversity and inclusion into their LG, KSE, and AFJ, as evidenced by a higher cumulative average score. Additional descriptive results of the current sample can be found in Table 1.

Results from the thematic criteria scores provide further insight into how FPCs are or could be more effective in embedding diversity and inclusion into their LG, KSE, and AFJ. Table 2 summarizes thematic scores from the cluster analysis of the FPCs in our selected cities. Our findings indicate that the FPCs scored relatively high and thus appear to be effective in embedding diversity and inclusion into their KSE (73%), LG (54%), and AFJ (49%) themes, respectively. Using the cumulative average for each theme, we group the FPCs as having “high” or “low” effectiveness. A total of five groups emerge from this analysis. There were four outliers: San Francisco, CA, USA; San Antonio, TX, USA; Indianapolis, IN, USA; Los Angeles, CA, USA. These outlier FPCs were grouped with FPCs that shared similar cumulative and thematic average results found in Table 2. Group A included FPCs representing Washington, DC, USA; Philadelphia, PA, USA; Austin, TX, USA; Denver, CO, USA; and Columbus, OH, USA. Group B included FPCs representing Boston, MA, USA; Chicago, IL, USA; San Diego, CA, USA; Fort Worth, TX, USA; and Los Angeles, CA, USA. Group C included FPCs representing San Jose, CA, USA; Houston, TX, USA; Seattle, WA, USA; Indianapolis, IN, USA; and San Francisco, CA, USA. Group D included FPCs representing New York, NY, USA; Phoenix, AZ, USA; Charlotte, NC, USA; San Antonio, TX, USA.

Group A was the most effective in advancing diversity and inclusion as evidenced by a “high” score in all three of the themes that we evaluated. While Group B and Group C resulted in only a 5% difference in their cumulative average score, there are major differences in their individual LG, KSE, and AFJ theme scores. In the LG theme, Group B scored 18%, while Group C scored 73%. Conversely, Group B scored higher in KSE and AFJ themes as compared to Group C. Group B and C scored 83% and 50% for KSE, respectively. Similarly, Group B and C scored 65% and 25% for the AFJ theme, respectively. Group D was the only group that scored “low” in all three themes and thus achieved the lowest cumulative average score. Additional group and thematic scores can be found in Table 2. Figure 3 displays results by theme to support the comparative analysis between the four groupings of urban FPCs.

### 3.2. Comparative Results of the Effectiveness of Urban FPCs

Results from this study identify different patterns among the urban FPCs and their scores in the KSE (73%), LG (54%), and AFJ (49%) objectives, respectively (Figure 3). Group A comprises the most effective FPCs with “high” scores in each theme: LG (90%), KSE (97%), and AFJ (73%). Figure 4 provides details of the average score of Group A for each individual theme. While Group A scored the highest amongst the urban FPCs, there is still an opportunity to improve on individual scoring criteria. For example, if the FPCs in Group A are committed to diversity and inclusion, including a visible purpose/mission statement on their website (AFJ.1) could improve their overall AFJ score. It is noteworthy that the five FPCs in Group A are all government-affiliated FPCs that were established through a local municipal government (mayoral/executive order or legislation). These results suggest that municipal governments are important stakeholders to promote greater transparency and accountability in all three of our diversity and inclusion measures.

Group D was the lowest scoring group with a cumulative average score of 38%. The 4 FPCs in this group scored “low” on all three measures: 31% in LG, 50% in KSE and 31% in AFJ. Figure 5 provides details on the average score of group D based on the individual criteria in the three diversity themes. Group D is diverse in that half of the FPCs are affiliated with and established through their local municipal government (New York, NY, USA; Phoenix, AZ, USA), while the other half is affiliated with a nonprofit organization and was established via a grant or other collaborative effort (Charlotte, NC, USA; San Antonio, TX, USA). There were no major differences in the average cumulative scores of these two subgroups of Group D. Comparative results of the LG and KSE themes reveal differences in effectiveness. Government-affiliated FPCs in Group D scored higher in the LG criteria (50% vs. 13%), while FPCs affiliated with a nonprofit organization scored higher in KSE (75% vs. 42%). These differences in the individual thematic scores are further supported by the differences in the performance of Group B and Group C.

Cumulative scores for Group B (53%) and Group C (48%) were relatively close. The descriptive details and theme scores between the two groups indicate that Group B is comprised of FPCs that are nonprofits, or were established by a local coalition. Group C consists of predominantly government-affiliated FPCs that were established by the local municipality via legislation or mayoral/executive order. On average, Group C (73%) scored higher in LG compared to Group B (18%). This was evidenced by the presence of governance documents/bylaws (LG.2), details about leadership objectives (LG.3), and diverse representation in the FPC leadership (LG.4) in their publicly available documents (Figure 6). These results suggest that municipal governments may have a higher priority than non-profits for publicizing the diversity efforts of the FPC in their jurisdiction. 

On the other hand, improvement of the overall criteria of FPCs in Group B can improve their LG and cumulative score (Figure 7). Group B scored higher and thus was more effective in embedding diversity and inclusion into their KSE and AFJ themes. In KSE, Group B scored an average of 83% while Group C only scored 50% (Figure 3). For the FPCs in Group C, the biggest opportunity for improvement in KSE includes displaying contact and/or public meeting information (KSE.1) and operating at least two modes of communication (e.g., listserv/newsletter, blog/news update page, social media, etc.) (KSE.2) (Figure 7). Additionally, FPCs in both Group B and Group C can improve their KSE and cumulative score by displaying essential website information (KSE.3), including their mission/vision, meeting notes, and accessible language on their website (Figure 7). In this study, only five urban FPC websites were accessible in languages other than English (New York, NY, USA; San Francisco, CA, USA; Denver, CO, USA; Seattle, WA, USA; Phoenix, AZ, USA). The absence of essential website information and other KSE criteria may hinder communication, engagement, and participation from residents, potential partners, and funders. These deficits in visibility impact the overall rating of the FPCs in this groups even as activities and community engagement efforts are reported elsewhere [3,46,47].

## 4. Discussion

This analysis provides an assessment of the diversity and inclusion related structures and activities of food policy councils in the cities in our sample. Our approach distinguishes two levels of analysis. Results from our national content analysis across our entire sample provide secondary level data that falls into five clusters. By examining trends and patterns among the identified groups of FPCs, we found considerable differences in their effectiveness in embedding diversity and inclusion into the leadership and governance (LG) structure, their key stakeholder engagement (KSE), and their activities toward food justice (AFJ).

Our results also reveal differences between the FPC affiliation with the local municipal government and cumulative FPC score (Table 2). This distinction is evident in both levels of analysis. The most common pathway of forming an FPC is through municipal governance via legislation or mayoral/executive order (52%). These FPCs resulted in a higher cumulative score (63%). Furthermore, government-affiliated urban FPCs (63%) scored higher than nonprofit urban FPCs (48%). These results suggest that the combination of grassroots advocacy, municipal governmental relationships and policy prioritization is a stronger indicator of FPC effectiveness. Additionally, urban-based FPCs that partner with the city and surrounding county score higher (62%), and thus, are more effective in embedding diversity and inclusion into their LG, KSE, and AFJ themes compared to urban FPCs representing the city (56%) or county (50%) separately (Table 2). These results align with existing research highlighting the importance of government relations to mobilize partnerships and operationalize food policy and food-related funds within a municipality or county [36,42,47].

Our results align with existing research regarding the effectiveness of municipal governments across the country in resourcing staff positions dedicated to food systems and policy [36,37]. Urban FPCs are commonly associated with a municipality’s office of sustainability, environment or health. These agencies are often responsible for spearheading intergovernmental and cross-sector collaborations. In some instances, municipal governments dedicate an entire office or team toward food policy (New York, NY, USA; Boston, MA, USA; Denver, CO, USA). Our sample includes three cities that have intergovernmental task forces and offices operating as their main food policy institution (New York, NY, USA; Seattle, WA, USA; Phoenix, AZ, USA). These FPCs are unlikely to maintain governance documents such as bylaws (LG.2) or details about leadership positions (LG.3) that deviate from existing government policies. The maximum attainable score for these cities was therefore adjusted to avoid distortions in our results.

We recognize that our methodology may not capture all details regarding FPC composition, priorities, and activities since it is based on publicly available information. We did not expressly collect data on our selected diversity themes. Still, our analysis provides critical insights into the perspective of the FPC website user—such as food consumers—whose perception about their local FPC will likely be based on publicly available information. Our study does not analyze the quality of FPC websites or of the documents we analyzed. For example, more than half of the urban FPCs in our sample did not have a visible commitment to diversity and inclusion in the form of a purpose/mission statement on their website (AFJ.1). The results from the AFJ theme highlight opportunities for deeper engagement, strategy, and support for both urban food producers and consumers through, for example, working group priorities (AFJ.2) and published reports about the work of the FPCs (AFJ.3). On the other hand, the maintenance of resources available to support both urban producers and consumers (AFJ.4) is one of the highest scoring criteria across all FPCs (87%). Follow up research may provide further insights regarding the actual priorities of FPCs versus those revealed in their public documents.

Further analysis may also be needed regarding FPC leadership positions. This study emphasizes the process of how FPC leaders are identified and selected rather than who those leaders are. For example, FPCs in Boston, MA, and Indianapolis, IN hold elections for council leadership positions. This is an example of how the pathway to leadership (LG.3) creates an opportunity to increase diversity and inclusion of perspectives within the leadership body, which may in turn be reflected in FPC activities. A limitation of our approach is that diversity is assessed via the representation of urban producers and consumers. However, this metric does not reveal the underlying representation of communities of color or other underserved and underrepresented groups. Resources are available to support FPCs in establishing best practices and engagement strategies that expressly engage underrepresented groups [15,59,60]. Future studies might assess the representation and inclusion of these groups given their unique experience within the urban food system and the systemic challenges that often limit or outright exclude them from accessing its benefits.

Our results suggest that future research might further assess the effectiveness of urban FPCs based on regional objectives and higher-level funding, including funding from the USDA. Due to our sample selection criteria, there is little representation of FPCs from the Midwest (12%), the largest agricultural-producing region in the U.S. Active FPCs in the Midwest may have experiences that can improve local food systems objectives, and broader economic and environment objectives that other FPCs might replicate. Half of the selected cities in our sample (*n* = 13) represent five of the top ten agricultural output producing states in the U.S. [26]. Collectively, California, Texas, Illinois, Indiana, and North Carolina produced $125B in agriculture commodities in 2020 [26]. These states also provided $31B to SNAP recipients in 2020 [61]. Future research might consider additional states to analyze additional themes for successfully aligning food systems leadership and desired food producer and food consumer benefits. Sharing best practices, challenges and innovations needs in view of the upcoming Farm Bill reauthorization in 2024 is important for FPCs to receive the resources they need to improve the representation of diverse perspectives of food systems stakeholders in their respective jurisdiction.

## 5. Conclusions

In 2022, the USDA announced initiatives designed “to achieve nutrition security through meaningful support, healthy food, collaborative action and equitable systems” including the recognition of urban agriculture and the implementation of the first-ever DEIA strategic plan [2,34]. To support the recent move toward more equitable food systems, this study provides a framework for food systems leaders, specifically in large urban communities, to assess the visibility and effectiveness of their diversity and inclusion measures as reflected by their leadership and governance (LG) structure, key stakeholder engagement strategies (KSE), and activities advancing food justice (AFJ). Results from this national assessment of 19 urban food policy councils reveal that they are effective in embedding diversity and inclusion in their KSE (73%), LG (54%), and AFJ (49%), respectively. A cluster analysis of the FPC thematic scores revealed four groups and highlights opportunities to improve individual, thematic, and cumulative scores. The cluster analysis also revealed descriptive and thematic trends in the distinct cluster groups. Groups A and C were predominantly composed of government-affiliated FPCs and scored higher in LG and in the average cumulative score. Conversely, Group B was predominantly comprised of FPCs categorized as nonprofit organizations and scored higher in KSE and AFJ as compared to Group C. Overall, urban FPCs are effective in providing resources to urban producers and consumers (AFJ.2). However, there remains an opportunity for urban FPCs to strengthen their overall AFJ score with a visible commitment to diversity and inclusion via their public facing communication tools, especially their websites, working group priorities, and published reports. Future research should continue to analyze FPC results by descriptive details, including organizational affiliation and region, while including new criteria to specifically assess representation from communities of color or other underserved and underrepresented groups. Participation from food systems leadership groups such as FPCs will be vital in advocating for equitable food systems through urban producer and consumer programs in the upcoming reauthorization of the U.S. Farm Bill in 2024.

## Figures and Tables

**Figure 1 foods-12-01854-f001:**
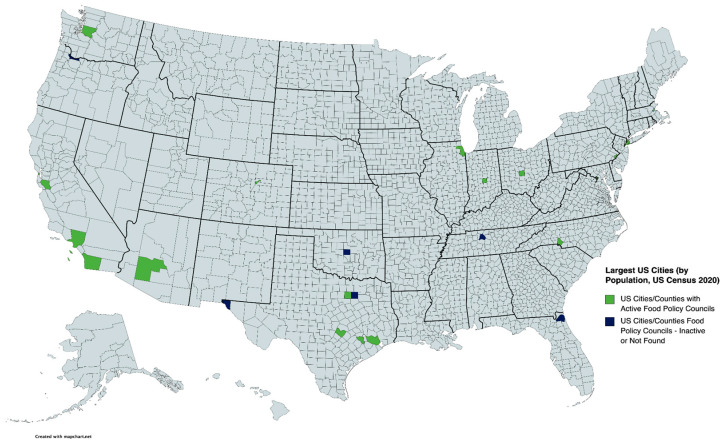
The 25 largest United States (U.S.) cities and the status of their respective food policy councils (FPCs). Created with www.mapchart.net.

**Figure 2 foods-12-01854-f002:**
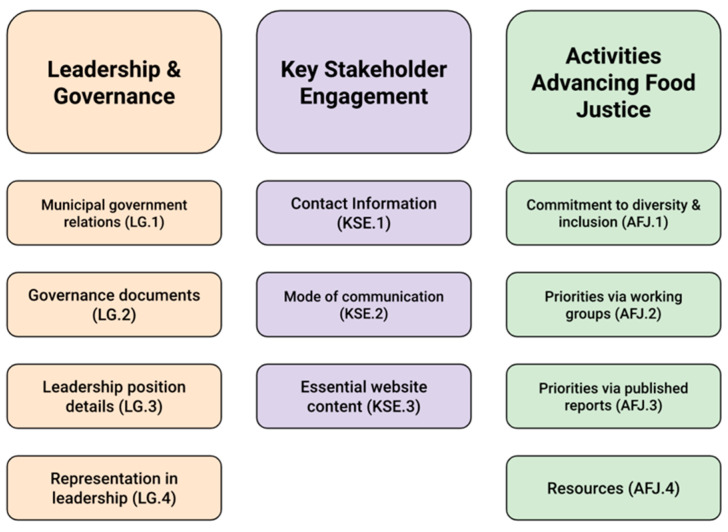
A framework to analyze the effectiveness of urban FPCs in the U.S.

**Figure 3 foods-12-01854-f003:**
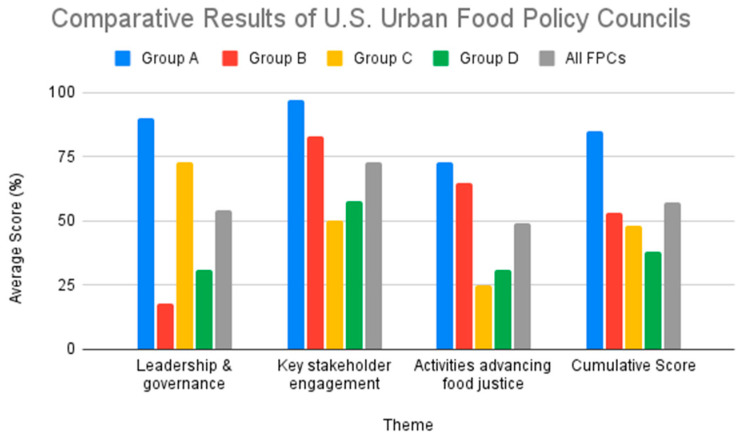
Comparative results of U.S. urban FPCs by theme.

**Figure 4 foods-12-01854-f004:**
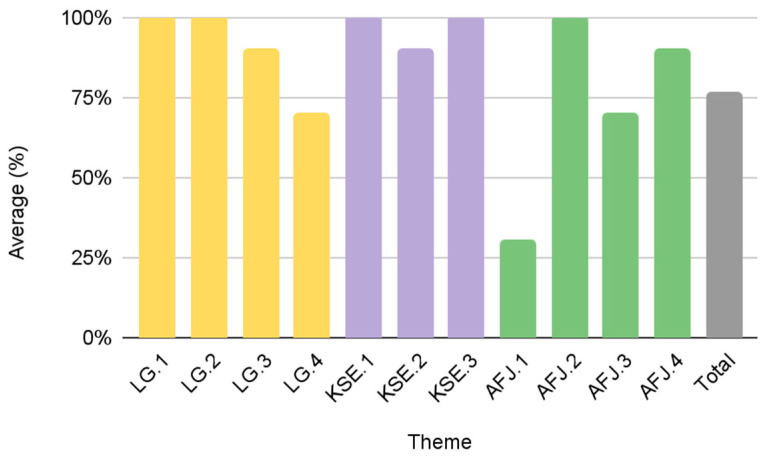
Group A average scores by theme criteria (*n* = 5).

**Figure 5 foods-12-01854-f005:**
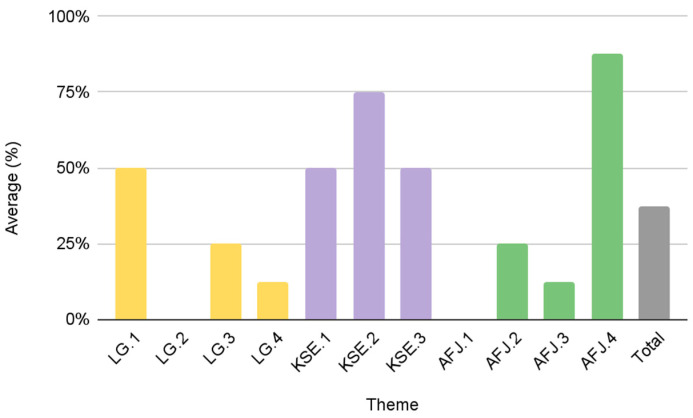
Group D average scores by theme criteria (*n* = 4).

**Figure 6 foods-12-01854-f006:**
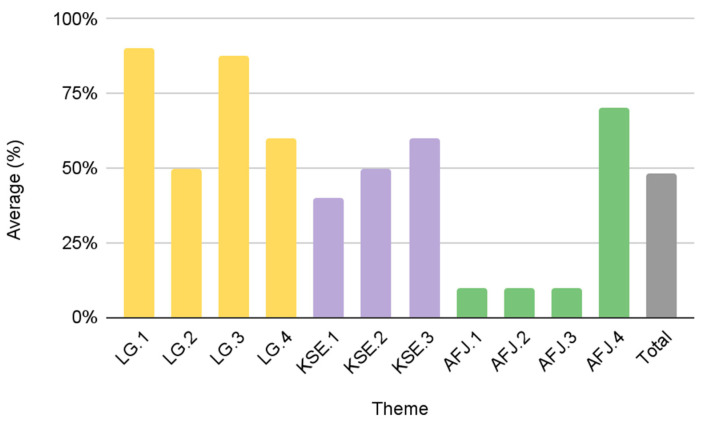
Group C average scores by theme criteria (*n* = 5).

**Figure 7 foods-12-01854-f007:**
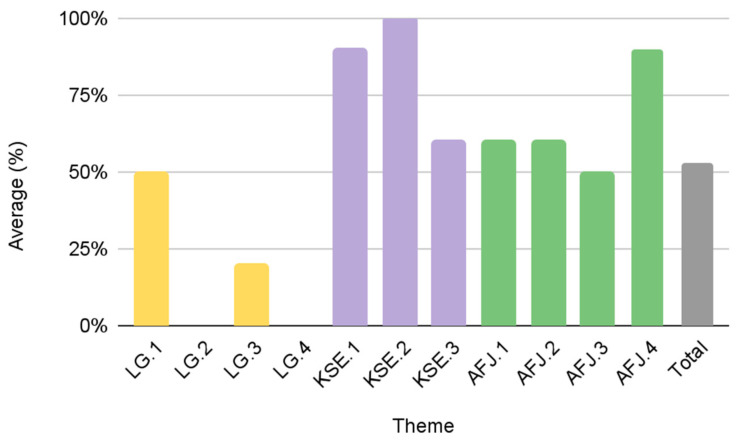
Group B average scores by theme criteria (*n* = 5).

**Table 1 foods-12-01854-t001:** Results of the Diversity and Inclusion Analysis of Food Policy Councils (FPCs) in Major Cities in the United States (U.S.)

Category	Type	N (%)	Cumulative Score (%)
Organizational Affiliation	Government advisory group, office of intergovernmental task force	14 (56%)	63
	Nonprofit organization 501(c)(3)	8 (32%)	48
	Coalition	0 (0%)	0
	N/A	3 (12%)	0
Locale Representation	City/Municipality	11 (44%)	56
	City–County	7 (28%)	62
	County	4 (16%)	50
	N/A	3 (12%)	0
Pathway of Inception	Municipality Government (Mayoral/ Executive Order; Municipal Legislation)	13 (52%)	63
	Coalition	5 (20%)	46
	Grant	4 (16%)	48
	N/A	3 (12%)	0
Region in the U.S.	West	8 (32%)	52
	South	11 (44%)	58
	Midwest	3 (12%)	59
	Northeast	3 (12%)	65

**Table 2 foods-12-01854-t002:** Comparative results of U.S. urban FPCs’ effectiveness in Leadership and Governance (LG), Key Stakeholder Engagement (KSE) and Activities advancing Food Justice (AFJ).

Group	Total (n)	FPCs in Group	LG (%)	KSE (%)	AFJ (%)	Cumulative Score (%)
All	19	-	54	73	49	57
Group A	5	Washington, DC, USA; Philadelphia, PA, USA; Austin, TX, USA; Denver, CO, USA; Columbus, OH, USA	90 (High)	97 (High)	73 (High)	85
Group B	5	Boston, MA, USA; Chicago, IL, USA; San Diego, CA, USA; Fort Worth, TX, USA; Los Angeles, CA, USA	18 (Low)	83 (High)	65 (High)	53
Group C	5	San Jose, CA, USA; Houston, TX, USA; Seattle, WA, USA; Indianapolis, IN, USA; San Francisco, CA, USA	73 (High)	50 (Low)	25 (Low)	48
Group D	4	New York, NY, USA; Phoenix, AZ, USA; Charlotte, NC, USA; San Antonio, TX, USA	31 (Low)	58 (Low)	31 (Low)	38

## Data Availability

All data sources are publicly available and referenced in the text.

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
