# Peer review of "Measuring the Effectiveness of Food Policy Councils in Major Cities in the United States"

_foods, 2023, doi:10.3390/foods12091854_

Round 1

Reviewer 1 Report

In my opinion, the authors submitted a manuscript that is interesting, and is linked to the objectives of the journal, however, there are some issues that have to be reconsidered. The Title is clear and express the objective of the manuscript.

The objective of the manuscript is to provide a framework for food systems leaders, specifically in large urban communities, to assess the visibility and effectiveness of their leadership & governance, key stakeholder engagement strategies (KSE) and activities advancing food justice.

The topic addressed could be, potentially, relevant for the journal and for the field, as large. The aim gap is pointed out in the studied literature review by exploring a lack of knowledge regarding the principles of competitive advantage.

The subject is rather interesting, so there is potential room for this manuscript, once it reaches the expected level of quality.   

The Abstract should be completed with some relevant findings (expressed in numbers, for a better impact on the reader). Also, in this part, a too-large share is allocated to the concept presentation compared with the result and conclusions. I would, also, advise, not to use abbreviations on the Abstract, but only on the body of the manuscript.

For better visibility on databases, the authors are asked not to repeat among keywords the words/concepts included in the title of the article. Entering different words in the title and in the keywords can improve the search for the paper in metasearch engines and internet databases.

In the introduction in fluent, but the presentation of the structure of the paper is missing. The main objective and the potential secondary objectives are sufficiently pointed out. 

The part of the Literature Review is totally missing. There are missing relevant information that are needed to construct the literature gap (similar studies on how the business performance of worldwide beekeeping production, in order to intensify and modernize production and use the potential of the competitive environment are totally missing). General text about the topic (one para), focus on its relevance and importance for the industry, practice, and theory. Then briefly inform readers that there is a surge in wellness experience, and xx and xx studies have been organized so far. One para on what has been done so far on the topic. Then talk about research gaps- What are the keys and why are they important or need to be addressed now? Clearly present 3-4 research gaps on this topic. Finally, present the focus of the current study. What are its RQs and details on the method – briefly say about data, country, context, and theory. Which is the novelty and contribution of the study?  

Research Methodology and Data Collection. It is recommended to present, as refences, similar studies, would improve the level of trust of the study.

The results are interesting, and looks consistent.

The discussion needs a short introductory paragraph explaining its structure could help better undemanding it.

The conclusions are short, easy to read and containing relevant information.

The references are not sufficient for designing the scientific gap

The tables are clear and enough. Figure 2 must be replaced, as it is not fully readable.

Author Response

Response to Comments and Suggestions by the Reviewers

April 21, 2023

Reviewer #1:

We thank the reviewer for the timely, encouraging, and helpful suggestions for our manuscript. 

We appreciate the generally positive feedback and have incorporated the reviewer’s comments

in the updated draft.

In response to the comments regarding the title, objectives, and topic of our manuscript. we removed “food systems leadership” from the title of the manuscript and added ‘leadership’ as a keyword. We also made some revisions to the abstract to address the reviewer’s comments regarding the use of abbreviations and we have highlighted our relevant findings. 

We appreciate the reviewer’s comments regarding the needs to further structure the introduction section of our manuscript. The following subtitles were added to the introduction section to add structure and clarity:

1.1 ​​Importance of diversity, equity and inclusion in the U.S. food system;

1.2. U.S. food policy – bridging production and consumption; 

1.3. Building inclusive and effective food systems leadership.

At the recommendation of the reviewer, we have provided additional context and relevant references to the manuscript to more adequately address the current state of knowledge about food policy councils and to add new knowledge that closes persistent knowledge gaps. Since urban food policy councils are novel institutions, and we hope our research will encourage additional new research in this field.

We have added additional evidence regarding the importance of diversity and inclusion as assessment criteria by drawing on the organizational and systems leadership literature. Examples from the private sector as well as recent examples from the public sector are now referenced to strengthen our argument regarding the importance of assessing the diversity and inclusion performance of the urban food sector. We draw on this related literature since urban food policy councils are novel institutions, and most of them are less than 10 years old. Research is therefore sparce. Given the history of the U.S. food systems and the lack of diversity in organizational leadership in the United States Department of Agriculture, we aim to add awareness and evidenced-based research regarding the importance of diversity and inclusion to the food sector as reflected in the structure, activities and impact of food policy councils in major urban cities.

We highlight the growing literature on diversity and inclusion in fields other than urban food systems, and have included assessments from other fields that confirm the importance of attention to diversity and inclusion in improving performance overall. As a result of this added review of the literature, 12 references were added to the manuscript to include relevant and sufficient evidence for our argument that we have identified an existing gap and are submitting a study to that begins to address it.

We appreciate the reviewer’s recommendation to improve the  information regarding our research methodology and data collection, we added several references and revised the text to include stronger evidence for the relevance of our approach in addressing existing research gaps. To address the reviewer’s recommendations, we have made two revisions.  One, we have added to  the  introduction of the paper and now point out that Food Policy Councils are on average less than ten years old and literature assessing their performance in general and diversity and inclusion in particular is sparce to date.

We also added  an introductory paragraph to the discussion section of the manuscript as suggested by the reviewer.

Finally, we have expanded our discussion of results and added several references comparing our results with the existing literature and implications for policy and practice.

Figure 2 was replaced with an updated and reformatted version to ensure readability.

Again, we thank the reviewer for the helpful comments and appreciate the opportunity to improve our manuscript based on the suggested revisions.

Respectfully,

Camille Range, Sabine O’Hara, Tia Jeffery and Etienne Toussaint

Reviewer 2 Report

This is a good paper

two revisions

1. add a research gap (right before the sentence starting with "This study assesses the effectiveness of FPCs...")

2. Your Discussion does not start until the paragraph starting with "This study emphasizes the process of how FPC leaders are identified and selected". Everything before that is Results or perhaps "Analysis"
Your Discussion should include sections that  (i)compare and contrast  your results with with literature and (ii) implications for policy, outreach, practice, etc.

also, Fig 2 is unreadable, please fix

Author Response

Thank the reviewer for the overall positive feedback. We made two major revisions to the manuscript as recommended by the reviewer. We have expanded information about the background and relevance of our research to the introduction of the paper and have also restructured the introduction for added clarity and readability.

Secondly, we have added 12 references to address the reviewer’s recommendations to add references regarding the current knowledge about food policy councils and the existing research gap. In addressing the reviewer’s recommendations, we drew on the literature in fields other than urban food systems. The reasons for drawing on other fields that confirm the importance of attention to diversity and inclusion in improving performance overall is that food policy councils are novel institutions and the majority are less than 10 years old. Research is therefore sparce.  .

To address the reviewer’s feedback regarding the discussion  section of our paper, we added subheadings in the results section to provide more clarity and structure to the discussion of our different findings.  In addition, we expanded the discussion section to include a comparison of our results with results from the existing literature and identify additional future implications. Current research results align with existing research of the importance and impact FPCs can have based on their affliction with local municipal governments.

We also conclude that future research is needed to consider the effectiveness of other food system leadership and organizational types, specifically those linked to municipal offices dedicated to food systems improvements and policies.

Finally, we updated figure 2 in the text and reformatted it to ensure readability. within the manuscript . The supplemental materials also include a powerpoint file of each figure from the manuscript.

Again, we thank the reviewer for the helpful comments and appreciate the opportunity to improve our manuscript based on the suggested revisions.

Respectfully,

Camille Range, Sabine O’Hara, Tia Jeffery and Etienne Toussaint
